# Status of the muEDM Experiment at PSI †

**Kim Siang Khaw** [1,*], **Cheng Chen** [1], **Massimo Giovannozzi** [2], **Tianqi Hu** [1], **Meng Lv** [1], **Jun Kai Ng** [1], **Angela Papa** [3,4,5], **Philipp Schmidt-Wellenburg** [3], **Bastiano Vitali** [4,6] **and Guan Ming Wong** [1] on behalf of the muEDM Collaboration

1    Tsung-Dao Lee Institute, School of Physics and Astronomy, Shanghai Jiao Tong University, Shanghai 201210, China; chencheng92@sjtu.edu.cn (C.C.); hutianqi@sjtu.edu.cn (T.H.); meng.lv@sjtu.edu.cn (M.L.); ngjunkai@sjtu.edu.cn (J.K.N.); wong.gm@sjtu.edu.cn (G.M.W.)
2    Beams Department, CERN, Esplanade des Particules 1, 1211 Meyrin, Switzerland; massimo.giovannozzi@cern.ch
3    Paul Scherrer Institut, Forschungsstrasse 111, 5232 Villigen, Switzerland; angela.papa@psi.ch (A.P.); philipp.schmidt-wellenburg@psi.ch (P.S.-W.)
4    Istituto Nazionale di Fisica Nucleare, Sez. di Pisa, Largo B. Pontecorvo 3, 56127 Pisa, Italy; bastiano.vitali@psi.ch
5    Dipartimento di Fisica, Università di Pisa, Largo B. Pontecorvo 3, 56127 Pisa, Italy
6    Dipartimento di Fisica, Università di Roma "La Sapienza", Piazzale Aldo Moro 2, 00185 Roma, Italy
*    Correspondence: kimsiang84@sjtu.edu.cn
†    Presented at the 23rd International Workshop on Neutrinos from Accelerators, Salt Lake City, UT, USA, 30–31 July 2022.

**Abstract:** Permanent electric dipole moments (EDMs) are excellent probes of physics beyond the Standard Model, especially on new sources of CP violation. The muon EDM has recently attracted significant attention due to discrepancies in the magnetic anomaly of the muon, as well as potential violations of lepton-flavor universality in B-meson decays. At the Paul Scherrer Institute in Switzerland, we have proposed a muon EDM search experiment employing the frozen-spin technique, where a radial electric field is exerted within a storage solenoid to cancel the muon's anomalous spin precession. Consequently, the EDM signal can be inferred from the upstream-downstream asymmetry of the decay positron count versus time. The experiment is planned to take place in two phases, anticipating an annual statistical sensitivity of $3 \times 10^{-21}$ $e \cdot$cm for Phase I and $6 \times 10^{-23}$ $e \cdot$cm for Phase II. Going beyond $10^{-21}$ $e \cdot$cm will enable us to probe various Standard Model extensions.

**Keywords:** electric dipole moment; frozen-spin technique; CP violation; lepton universality

## 1. Introduction

The origin of the imbalance between baryon and anti-baryon in our universe (BAU) [1] remains one of the greatest mysteries in cosmology and particle physics. The size of the Charge-Parity (CP) symmetry violation embedded in the Standard Model (SM) of particle physics is insufficient to explain the observed BAU [2]. The existence of a permanent electric dipole moment (EDM) in any elementary particle inherently violates both the P and T symmetries [3]. Assuming CPT symmetry, the latter implies a violation of the CP symmetry. If an EDM measurement exceeds the SM prediction, it could suggest physics beyond the Standard Model (BSM) [4–7], thus refining our understanding of the universe.

Recently, muon EDM has drawn considerable attention due to discrepancies between the magnetic anomaly of the muon $a_\mu$ [8–10] and the electron $a_e$ [11–13], along with suggestions of lepton-flavor universality violation in B-meson decay [14]. Of all elementary particles, only the muon permits a direct EDM measurement. The present muon EDM limit, set by the BNL Muon $g - 2$ Collaboration, is $d_\mu < 1.8 \times 10^{-19}$, $e \cdot$cm at a 95% confidence level [15], while the SM prediction is $1.4 \times 10^{-38}$ $e \cdot$cm [16,17], way beyond the reach of current technology. Consequently, the muon EDM remains one of the SM's least tested areas, with any detected signal strongly implying BSM physics.

The current muon EDM limit is derived from a "parasitic" measurement within the Muon $g - 2$ experiment. In this context, the existence of the muon EDM induces a tilt $\delta = \tan^{-1}(\eta\beta/2a_\mu)$ in the $g - 2$ precession plane, where $\eta$ is a dimensionless parameter related to muon EDM and $|\vec{d}_\mu| \approx \eta \times 4.7 \times 10^{-14}$. The current BNL limit implies a plane tilt $\delta$ around a milliradian, corresponding to an average vertical angle oscillation for emitted positrons of several tens of µrad. The sensitivity projected for the FNAL and J-PARC experiments is around the order of $10^{-21}$ $e$·cm [18,19].

## 2. A Frozen-Spin-Based Muon EDM Search at PSI

The recently proposed frozen-spin technique [20–22] enhances sensitivity in an EDM search by canceling the muon's anomalous spin precession in a storage magnet using a radial electric field $E_f = a_\mu Bc\beta\gamma^2$, where $B$ is the B-field confining the muon. This technique effectively "freezes" the muon's spin in relation to its momentum. If the muon has a permanent EDM, it would induce a spin rotation out of the orbital plane, leading to an observable upstream-downstream asymmetry that increases over time. By placing detectors in the upstream and downstream direction, an up-down asymmetry $\alpha$ in positron count can be observed due to increasing polarization along the direction perpendicular to the muon orbital plane, as the positron is preferentially emitted in the direction of the muon's spin. The sensitivity of the EDM measurement can then be calculated using:

$$\sigma(d_\mu) = \frac{\hbar\gamma^2 a_\mu}{2PE_f\sqrt{N}\gamma\tau_\mu\alpha} \tag{1}$$

where $P$ is the spin polarization, $N$ is the number of positrons, and $\tau_\mu$ is the muon lifetime.

The proposed muEDM experiment at PSI is planned in two stages. Phase I will employ an existing PSC solenoid with a field strength of 3 T to demonstrate the essential techniques required for a muon EDM search using the frozen-spin method. As illustrated in Figure 1, a surface muon beam with approximately 30 MeV/c momentum and a polarization greater than 95% will be directed into a collimation tube within a superconducting magnetic shield, creating a field-free condition. The use of correction coils within the solenoid will minimize the field gradient between the injection region at the collimation tube's exit and the storage region at the solenoid's center, thereby expanding the acceptable phase space. A coil situated at the solenoid's center will generate a weakly focusing field essential for muon storage during the EDM measurement.

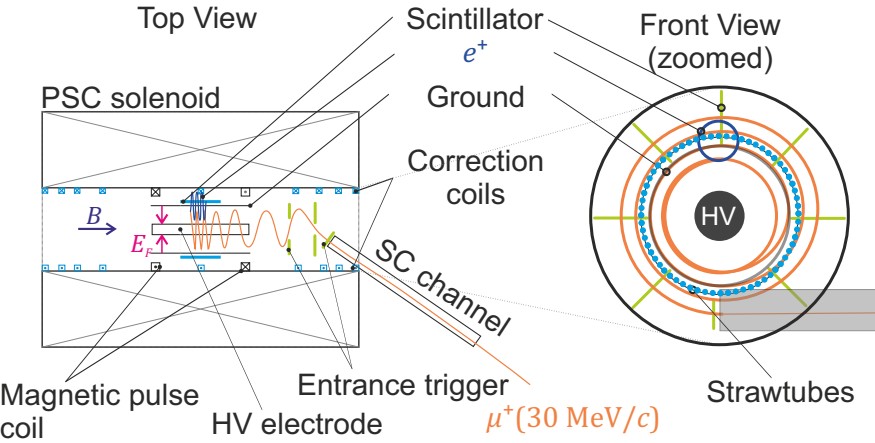

**Figure 1.** A sketch of the Phase I muEDM experiment at PSI. Side view (**left**) and front view (**right**) of the demonstrator device using an existing solenoid magnet with a field strength of 3 T.

An entrance scintillator, operating in anti-coincidence with a set of scintillators, will produce an entrance signal for muons within the acceptance phase space. This signal will trigger a pulsed magnetic field at the solenoid's center, converting the remaining

longitudinal momentum of the incoming muons into the transverse direction. The muon will then be confined to a stable orbit of approximately 30 mm radius within the weakly focusing magnetic field. To establish the frozen-spin condition, a radial electric field of 3 kV/cm will be applied in the storage orbit region between two co-axial electrodes.

The experiment will employ a combination of silicon strip and scintillating fiber ribbon detectors to track the decay positron, enabling the measurement of the muon's anomalous precession frequency, $\omega_a$. This measurement will serve as a sensitive probe of the magnetic field. Furthermore, by plotting $\omega_a$ against the applied electric field and interpolating it to $\omega_a(E) = 0$, we can adjust the electric field to achieve the frozen-spin condition. This procedure also enables us to measure the up-down asymmetry.

In Phase II of the experiment, we plan to employ a muon beam with a higher momentum of 125 MeV/c and a larger radial electric field of 20 kV/cm. Given that the anticipated muon orbit will be significantly larger than in Phase I, we will design a dedicated solenoid magnet specifically for this phase. A comparison of the conditions in Phases I and II can be found in Table 1.

**Table 1.** Annual statistical sensitivity of the muon EDM measurement for muEDM Phases I and II.

| Parameters | muEDM Phase I | | muEDM Phase II | |
|---|---|---|---|---|
| | Factor | Event Rate (Hz) | Factor | Event Rate (Hz) |
| Muon flux ($\mu^+$/s) | - | $4 \times 10^6$ | - | $1.2 \times 10^8$ |
| Channel transmission | 0.03 | $1.2 \times 10^5$ | 0.005 | $6 \times 10^5$ |
| Injection efficiency | 0.017 | $2.0 \times 10^3$ | 0.6 | $3.6 \times 10^5$ |
| $e^+$ detection efficiency | 0.25 | $5.0 \times 10^2$ | 0.25 | $9.0 \times 10^4$ |
| Detected $e^+$ per 200 days | $8.64 \times 10^9$ | | $1.56 \times 10^{12}$ | |
| Beam momentum (MeV/c) | $\approx$30 | | 125 | |
| Gamma factor, $\gamma$ | 1.04 | | 1.56 | |
| Storage magnetic field, $B$ (T) | 3 | | 3 | |
| Electric field, $E_f$ (kV/cm) | 3 | | 20 | |
| Muon decay asymmetry, $\alpha$ | 0.3 | | 0.3 | |
| Initial polarization, $P_0$ | 0.95 | | 0.95 | |
| **Muon EDM Sensitivity** ($e\cdot$cm) | $3 \times 10^{-21}$ | | $6 \times 10^{-23}$ | |

## 3. R&D Progress at PSI

From 2019 to 2022, we conducted four comprehensive test-beam measurements at PSI. In 2019, our primary focus was characterizing the phase space of potential muon beamlines. In 2020, we primarily studied the effects of multiple Coulomb scattering of low-momentum positrons. We reported the details from these test-beam measurements in [23,24]. Currently, we are analyzing the data from the 2021 testbeam, where we characterized potential electrode materials with positrons and muons. In 2022, we tested a prototype of the muon tagger/tracker, consisting of a time projection chamber (TPC), and then tested muon entrance detector prototypes using a 27.5 MeV/c muon beam from the $\pi$E1 beamline.

In Figure 2, a muon entrance detector based on a thin plastic scintillator (gate detector) and four GNKD [25] plastic scintillating tiles (telescope detectors), each readout by NDL [26] silicon photomultipliers (SiPMs) NDL15-6060S, was developed. The SiPMs were coupled to the scintillators using BC-603 optical grease. The scintillators and readout electronics were held together by a 3D-printed holder and a rack made of resin material. This prototype was recently tested at the $\pi$E1 beamline at PSI. Signal digitization was performed using WaveDAQ [27] based on DRS4 [28] (see Figure 3). A total of 0.7 million muon events were collected during the test beam, and data analysis is ongoing to characterize the efficiency of the muon entrance detector.

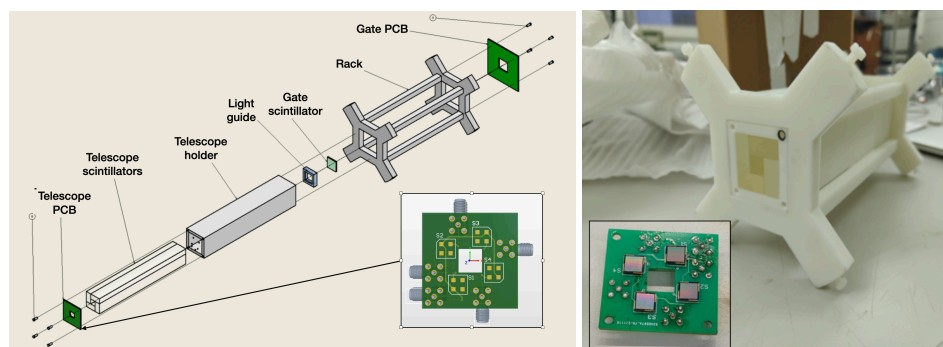

**Figure 2. Left**: A detailed three-dimensional CAD model of the muon entrance detector. **Right**: An image of the actual prototype entrance detector developed for the test beam at PSI. The close-up views (insets) display the SiPM readout boards designed for the telescope scintillators.

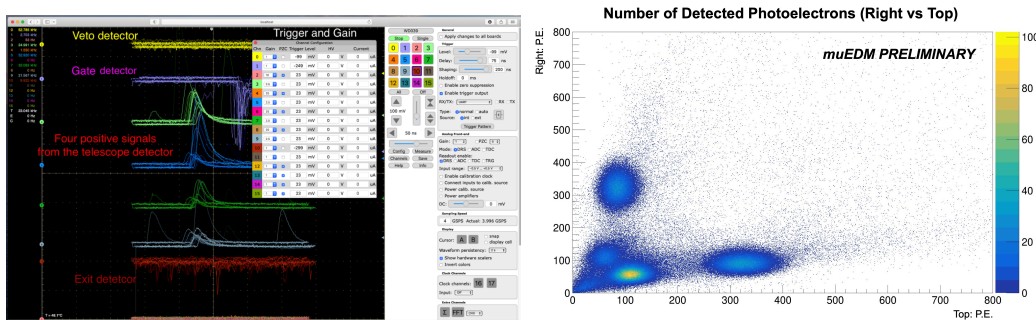

**Figure 3. Left**: Event display from the WaveDAQ digitizer. **Right**: Correlation plot between the top and right scintillators of the prototype entrance detector.

## 4. Conclusions and Outlook

A sensitive search of the muon EDM presents an exciting opportunity to explore BSM physics within the muon sector. The frozen-spin technique has the potential to surpass the sensitivity of current muon $g - 2$ storage ring experiments. By leveraging existing beamlines at PSI, it is possible to reach an unprecedented EDM sensitivity of $6 \times 10^{-23}$ $e \cdot$cm. Currently, the development and testing of the critical components for the experimental setup are in progress, setting the stage for the Phase I precursor experiment at PSI.

**Author Contributions:** Conceptualization, K.S.K. and A.P.; methodology, K.S.K., A.P. and P.S.-W.; software, T.H., G.M.W. and A.P.; hardware and electronics, M.L., T.H., J.K.N. and C.C.; validation, K.S.K., A.P. and P.S.-W.; formal analysis, T.H., G.M.W. and B.V.; investigation, T.H., J.K.N. and B.V.; resources, K.S.K., A.P. and P.S.-W.; data curation, A.P.; writing—original draft preparation, K.S.K.; writing—review and editing, M.G. and K.S.K.; visualization, J.K.N.; supervision, K.S.K., A.P. and P.S.-W.; project administration, K.S.K., A.P. and P.S.-W.; funding acquisition, K.S.K. and A.P. All authors have read and agreed to the published version of the manuscript.

**Funding:** This author was funded by the National Natural Science Foundation of China under Grant Nos. 12075151 and 12050410233.

**Institutional Review Board Statement:** Not applicable.

**Informed Consent Statement:** Not applicable.

**Data Availability Statement:** Not applicable.

**Acknowledgments:** We would like to thank our colleagues in the muEDM collaboration for useful discussions regarding the manuscript.

**Conflicts of Interest:** The authors declare no conflict of interest.

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
