# Peer review of "Status of the muEDM Experiment at PSI"

_psf, doi:10.3390/psf8010050_

Round 1

Reviewer 1 Report

Explained the importance of research about muon electric dipole moments also the frozen-spin application progress in PSI. But lack of results of experiment in the article(maybe because it is not in used yet). It would be good to see the data collected from the experiment and to perform some analysis.

Author Response

We thank the Editor and the reviewer for careful reading of the manuscript, and appreciate the valuable comments. We have made revisions based on your comments, as described below.

At the time of writing the manuscript, we have just completed the data taking and hence the results were not available. The first measurement of the muon EDM using this technique is yet to be realized as we are currently in the R&D stage of the experiment. 

The complete result from the test beam at PSI in 2022 will be published as a separate technical paper in another journal. We have included some preliminary results in the form of figure 3 for this conference proceeding. 

We have also added a recent publication regarding BSM physics related to Muon EDM in reference [4] to keep the paper up to date.